# Mortality Prediction Model before Surgery for Acute Mesenteric Infarction: A Population-Based Study

**DOI:** 10.3390/jcm11195937

**Published:** 2022-10-08

**Authors:** Shang-Wei Lin, Chung-Yen Chen, Yu-Chieh Su, Kun-Ta Wu, Po-Chin Yu, Yung-Chieh Yen, Jian-Han Chen

**Affiliations:** 1Department of Surgery, E-Da Hospital, Kaohsiung 82445, Taiwan; 2Healthcare Group Department of Medical Education, E-Da Hospital, Kaohsiung 82445, Taiwan; 3School of Medicine, College of Medicine, I-Shou University, Kaohsiung 82445, Taiwan; 4Division of General Surgery, E-Da Hospital, Kaohsiung 82445, Taiwan; 5Bariatric and Metabolism International Surgery Center, E-Da Hospital, Kaohsiung 82445, Taiwan; 6Division of Hematology-Oncology, E-Da Hospital, Kaohsiung 82445, Taiwan; 7Department of Psychiatry, E-Da Hospital, Kaohsiung 82445, Taiwan

**Keywords:** acute mesenteric infarction, surgery, bowel resection, preoperative risk factors, scoring system, mortality

## Abstract

Surgery for acute mesenteric infarction (AMI) is associated with high mortality. This study aimed to generate a mortality prediction model to predict the 30-day mortality of surgery for AMI. We included patients ≥18 years who received bowel resection in treating AMI and randomly divided into the derivation and validation groups. After multivariable analysis, the ‘Surgery for acute mesenteric infarction mortality score’ (SAMIMS) system was generated and was including age >62-year-old (3 points), hemodialysis (2 points), congestive heart failure (1 point), peptic ulcer disease (1 point), diabetes (1 point), cerebrovascular disease (1 point), and severe liver disease (4 points). The 30-day-mortality rates in the derivation group were 4.4%, 13.4%, 24.5%, and 32.5% among very low (0 point), low (1–3 point(s)), intermediate (4–6 points), and high (7–13 points)-risk patients. Compared to the very-low-risk group, the low-risk (OR = 3.332), intermediate-risk (OR = 7.004), and high-risk groups (OR = 10.410, *p* < 0.001) exhibited higher odds of 30-day mortality. We identified similar results in the validation group. The areas under the ROC curve were 0.677 and 0.696 in the derivation and validation groups. Our prediction model, SAMIMS, allowed for the stratification of the patients’ 30-day-mortality risk of surgery for acute mesenteric infarction.

## 1. Introduction

Despite significant improvements in surgical techniques and perioperative care, surgery performed in patients with acute mesenteric infarction (AMI) remains associated with high mortality [1,2]. Advanced mesenteric ischemia requiring bowel resection is associated with a 15-fold increase in mortality compared to that not requiring bowel resection [3,4]. Adaba et al. carried out a meta-analysis of observational studies in patients with AMI and observed that the pooled in-hospital mortality was 63% [5]. The operative mortality of mesenteric ischemia ranged from 26 to 72% with a pooled mortality rate of 47% (95% confidence interval (CI) = 40% to 54%) [6]. AMI is associated with high rates of major morbidities that vary depending on the type of surgical approach (such as ischemic bowel requiring resection, open revascularization, thromboembolectomy, and bypass) [7,8,9]. Recent publications describing mortality following bowel resection have indicated that most deaths occur within 30 days of the procedure [3,10]. Studies performed to date reveal that 30-day mortality is a reliable metric to determine the outcomes of surgery to treat AMI. Preoperative factors associated with 30-day mortality include patient vital parameters, comorbidities, clinical characteristics, duration between diagnosis and surgery, and intraoperative interventions [3,11,12,13,14,15,16,17,18,19,20,21,22,23,24,25].

Preoperative risk factors for predicting postoperative 30-day mortality in patients with AMI are not apparent because of the variations in design and investigated risk factors in previous studies; therefore, improving the prediction of such mortality remains an unmet need. Preoperative prediction is essential to determine whether patients need to undergo a bowel resection to assess intestinal viability and determine the feasibility of reestablishing blood supply to the ischemic bowel, removing the necrotic intestine, and promoting a care-related quality-of-life improvement process. Previous studies have identified some preoperative risk factors for 30-day surgical mortality post bowel resection [3,11,12,13,14,15,16,17,18,19,20,21,22,23,24,25]. However, few studies have involved scoring systems for predicting mortality following bowel resection, specifically among patients with AMI [19,20]. The clinical applications of such scoring systems remain limited because of varying preoperative risk factors (including the absence of some critical variables), the lack of evidence-based guidelines and proper validation studies [20,21]. The main limitation of preexisting scoring systems is a lack of impact due to the small number of studies performed.

Our study aimed to generate a mortality prediction model based on preoperative risk factors to predict the 30-day-mortality of surgery for AMI by analyzing data from the national database, NHIRD. The National Health Insurance (NHI) Research Database (NHIRD) of Taiwan is a large, powerful data source for biomedical research and a complete prospectively populated, multicenter repository of patient information; it includes all data available from primary outpatient departments and inpatient hospital care settings since the year 2000 [26]. We expected the model to provide evidence from a relatively large number of patients and to be easily accessible to clinicians in predicting surgical mortality outcome in treating acute mesenteric infarction.

## 2. Materials and Methods

### 2.1. Database and Study Sample

The study protocol was reviewed and approved by the Institutional Review Board of the E-da Hospital (No. EMRP-106-063). Informed consent is not required for this type of study. We used data from Taiwan’s NHIRD (registration number NHIRD-103-246), provided by the NHI Administration and the Ministry of Health and Welfare. We extracted data on inpatient expenditures by admission and information from the registry of beneficiaries and Registry of Catastrophic Illness Patient Database (RCIPD) entered into the NHIRD between 1 January 1996 and 31 December 2013.

### 2.2. Data Extraction and Inclusion/Exclusion Criteria

We extracted admission data of adult patients (≥18 years) who were admitted with mesenteric infarction between 2003 and 2012 from the NHIRD. Mesenteric infarction was defined based on the International Classification of Diseases-9th revision diagnostic code (ICD-9) 557.x, vascular insufficiency of the intestine. Bowel resection was defined by ICD-9: 45.6, resection of the small bowel [7,8,15].

Patients with chronic vascular insufficiency of the intestine only (ICD-9: 557.1), without bowel resection, and those of undetermined sex, were also excluded. Finally, 4897 patients were included and were randomly divided into the model derivation (*n* = 3918; 80%) and validation groups (*n* = 979; 20%) [27]. The study schematic is shown in Figure 1.

### 2.3. Primary Endpoint

The primary endpoint was all-cause mortality within 30 days, which was identified by discharge condition or removal from the NHI program [28]. The patients were either followed up until death or censored at the end of the study period (31 December 2013).

### 2.4. Covariant Assessment

The patients’ general conditions were evaluated based on their characteristics and comorbidities. After referring to previous studies, age, sex, Charlson Comorbidity Index score, and cardiac conduction disease were identified as the preoperative risk factors, and they were included in our study. Based on the details of the Charlson Comorbidity Index score, we chose 13 diseases to evaluate [3,11,12,13,14,15,16,17,18,19,20,21,22,23,24,25] (Appendix A). The cutoff value for continuous variables was determined using receiver operating characteristics (ROC) curves. Any admission records in the NHIRD that predated the index admission were used to identify the patients’ comorbidities as defined based on the Charlson comorbidity index [29]. Comorbidities recorded in the RCIPD of the NHIRD including dialysis, dementia, rheumatoid disease, malignancy, severe liver disease (defined as cirrhosis with intractable ascites, hepatic encephalopathy, or esophageal variceal bleeding), were also extracted. Other comorbidities such as hypertension (ICD-9: 401–405), hyperlipidemia (571.2, 571.4–571.6, 572.2–572.8, and 456.0–456.21), gout (582, 583, 585, 586, and 588), obesity (278.0x and 278.1x), and heart conduction disease (426.x and 427.x) [30,31,32,33] were included as covariates. 

### 2.5. Statistical Analysis

We used MySQL software to extract the data from our database and do randomization. Descriptive statistics and contingency tables were analyzed using the SPSS software version 22 (IBM Corp., Armonk, NY, USA). Differences in categorical variables in our model’s derivation and validation groups were analyzed using the Chi-square or Fisher’s exact test; such variables included age, sex, and comorbidities and were presented as numbers (percentages). Continuous covariates, including age and length of hospital stay, were analyzed using the Kolmogorov–Smirnov test and compared using the Wilcoxon rank-sum test for non-normal distributions, which are presented as medians (interquartile ranges) or using Student’s *t*-test for normal distributions, which are presented as means (standard deviations). A 2-sided *p*-value <0.05 was considered statistically significant.

For predicting 30-day mortality, all variables were analyzed using a univariate logistic regression model; factors with a modest (*p* < 0.2) association with 30-day mortality were included in a multivariable backward stepwise logistic regression model to calculate the odds ratio and regression coefficient. The regression coefficient of variables significantly related to mortality (*p* < 0.05) was multiplied by numbers and rounded to the nearest integer to create a score on an additive scale [4,34,35,36]; this score was applied to the model’s derivation group. The reliability of the predictive model was assessed with respect to discrimination using the area under the ROC curve (AUC) and calibration using a Hosmer–Lemeshow goodness-of-fit test. Finally, we applied the scoring system to the validation group to evaluate the model’s performance. The 30-day mortality rates across risk groups in both cohorts were compared by calculating the odds ratio (OR) of death within 30 days.

## 3. Results

### 3.1. Baseline Characteristics

Table 1 summarizes the characteristics of the 4897 patients in both the derivation (*n* = 3918) and validation (*n* = 979) groups. There were no significant differences between the groups regarding age, sex, or any of the covariates. The 30-day mortality rates in both groups were similar (17.46% vs. 17.16%, *p* = 0.851), as were the lengths of hospital stay (medians of 16 days for both, *p* = 0.318).

### 3.2. Generating the Scoring System for the Prediction Model

Multivariable analysis revealed that an age ≥62 years (adjusted OR 2.275, *p* < 0.001), as well as having a history of congestive heart failure (adjusted OR 1.475, *p* < 0.001), cerebrovascular disease (adjusted OR 1.275, *p* = 0.019), peptic ulcer disease (adjusted OR 1.513, *p* < 0.001), severe liver disease (adjusted OR 3.140, *p* < 0.001), hemodialysis (adjusted OR 1.736, *p* < 0.001), and diabetes (adjusted OR 1.353, *p* = 0.002) were associated with 30-day mortality (Appendix A). We tested several scoring systems to obtain the scoring system with the best reliability and calibration (Appendix A). The ‘Surgery for acute mesenteric infarction mortality score’ (SAMIMS) system was generated for these patients based on each of the factors mentioned earlier, including age >62-year-old (3 points), hemodialysis (2 points), congestive heart failure (1 point), peptic ulcer disease (1 point), diabetes (1 point), cerebrovascular disease (1 point), and severe liver disease (4 points) (Table 2).

### 3.3. The ‘Surgery for Acute Mesenteric Infarction Mortality Score’ (SAMIMS)

The distribution of the patients in the derivation and validation groups after applying the scoring system is presented in Appendix A. The cutoff was applied to 4 different homogeneous groups of risk according to their score: very low risk (0 points), low risk (1–3 points), intermediate risk (4–6 points), and high risk (7–13 points).

The distribution of patients according to their scores is reported in (Appendix A). The patient distributions according to 30-day mortality were similar across both groups. The AUCs were 0.677 (95% CI: 0.660–0.689, *p* < 0.001) and 0.696 (95% CI: 0.666–0.725, *p* < 0.001) in the derivation and validation groups, respectively. The model demonstrated good calibration per the Hosmer–Lemeshow χ^2^ test (Table 3). The 30-day mortality rates (Figure 2) in the derivation group were 4.4%, 13.4%, 24.5%, and 32.5% for very-low-, low-, intermediate-, and high-risk patients. Compared to the very low-risk group, the low- (OR 3.332, *p* < 0.001), intermediate- (OR 7.004, *p* < 0.001), and high-risk groups (OR 10.410, *p* < 0.001) demonstrated higher odds of 30-day mortality (Table 3). In the validation cohort, the 30-day mortality rates (Figure 2) were 3.5% for very-low-risk, 12.9% for low-risk, 24.7% for intermediate-risk, and 33.8% for high-risk patients. Compared to the very-low-risk group, the low-risk (OR 4.117, *p* = 0.001), intermediate-risk (OR 9.082, *p* < 0.001), and high-risk groups (OR 14.165, *p* < 0.001) demonstrated higher odds of 30-day mortality (Table 3).

## 4. Discussion

We established a scoring system, the ‘Surgery for acute mesenteric infarction mortality score’ (SAMIMS), to evaluate the risks of 30-day hospital mortality among patients who experienced mesenteric infarction and underwent bowel resection. The system included age >62-year-old (3 points), hemodialysis (2 points), congestive heart failure (1 point), peptic ulcer disease (1 point), diabetes (1 point), cerebrovascular disease (1 point), and severe liver disease (4 points). The 30-day mortality rates in the derivation group were 4.4%, 13.4%, 24.5%, and 32.5% among very-low- (0 point), low- (1–3 point(s)), intermediate- (4–6 points), and high-risk (7–13 points) patients. Compared to the very-low-risk group, the low-risk (odds ratio 3.332, *p* < 0.001), intermediate-risk (7.004, *p* < 0.001), and high-risk groups (10.410, *p* < 0.001) exhibited higher odds of 30-day mortality. We identified similar results in the validation group. The areas under the receiver operating characteristic curve were 0.677 (95% CI: 0.660–0.689) and 0.696 (0.666–0.725) in the derivation and validation groups, respectively. With our study analyzing data from more than 4000 patients, SAMIMS can enhance risk assessment and is easy to apply before surgery in daily practice.

It has been concluded that the management of specific etiologies of acute or chronic intestinal ischemia depends on the specific etiology (i.e., arterial occlusion or thrombosis, mesenteric venous thrombosis, and nonocclusive mesenteric ischemia). So, the standard treatments of occlusive and non-occlusive have been established. However, the management in different hospitals varies because of different patients’ conditions, the equipment and the health care staff. The hospitals with inadequate resources would not distinguish the etiologies of the AMI and treat the patients as standard guidelines. The patients in severe conditions or in the hospital with insufficient and inadequate medical equipment to follow treatment guidelines are thought to be the candidates to need bowel resection assessed by clinicians. Therefore, it is necessary to understand the prognosis of AMI after bowel resection. There are many different preoperative models for predicting mortality such as NELA score, SORT, SMPM, and POSPOM [37,38,39,40,41]. All of them are not specific to AMI after bowel resection.

We observed a postoperative 17% in-hospital mortality of AMI in our study. In contrast, the POSPOM study reported a relatively lower postoperative mortality rate of 0.5%. Several large-scale studies have reported postoperative mortality rates ranging from 1.3 to 4%. These scales did not totally fit the AMI patients. Compared to SAMIMS, several mentioned preoperative risk factors in POSPOM including ischemic heart disease, cardiac arrhythmia or heart blocks, peripheral vascular disease, dementia, hemiplegia, chronic obstructive pulmonary disease, chronic respiratory failure, chronic alcohol abuse, and cancer in POSPOM are not included in SAMIMS, which were designed for specifically predicting the 30-day mortality after bowel resection for AMI. It might be inaccurate if the patients had some risk factors unrelated to the postoperative mortality in AMI in other scoring systems. Thus, we believe the SAMIMS could provide more accurate results for the clinicians and help health care agent to understand.

A patient with 0–6 SAMIMS points is estimated to have a 4.4–24.5% probability of death; this rises to 32.5% with a score of 7–13 points. However, the mortality rates in this study were lower than those reported in a previous systemic review, where the short-term average mortality rate was 47% [6]. An explanation for the lower mortality rate in our study may be due to the easy accessibility of hospitals, doctors’ offices, clinics, and other health care providers in Taiwan. Moreover, the National Health Insurance in Taiwan allows patients to go and seek urgent medical attention when needed willingly. Another explanation could be a selection bias; surgeons might have chosen patients whose conditions were relatively stable and would likely have higher chances of survival after bowel resection. Therefore, our scoring system is most useful for surgeons to predict the surgical mortality outcome of patients receiving bowel resection due to AMI. The SAMIMS could thus help improve the understanding of the mortality risk for patients and their families before surgery.

Age is a significant factor in decision making and risk assessment. Our study demonstrated that an age ≥62 years is a significant predictor of postoperative death within 30 days (worth 3 points). Older patients deemed eligible for surgical treatment may not exhibit elevated mortality rates, but they still represent a high-risk population that requires comprehensive assessment and optimized treatment [11,42]. Another reason for old age as a significant risk factor is those elderly individuals with AMI exhibit atypical symptoms, which may lead to delayed diagnosis. Like previous reports, our analyses revealed that 3 points for age ≥62 years per the SAMIMS predicted at least a mortality rate of 13.4% post-laparotomy.

In our study, severe liver diseases had a strong negative impact on 30-day mortality (4 points); therefore, patients with severe liver disease should be treated cautiously. However, we posit that bowel resection can be performed safely in patients with milder disease, such as those with Child–Pugh, a stage with normal liver function test results and no ascitic edematous decompensation. Moreover, elevated hepatic aspartate aminotransferase (AST) is a preoperative risk factor as reported by Haga et al. and Merle et al. [20,21]; however, elevated AST likely indicates the degree of intestinal ischemia; its elevation alone without other indices related to liver function being affected does not strongly correspond to severe hepatic disease. The ‘Model for End-stage Liver Disease’ score has been suggested as an alternative to the Child–Pugh classification that provides for more appropriate risk assessment in patients with cirrhosis; however, this warrants further analysis [43,44,45,46]. Moreover, the patient with AMI is usually an emergency event. There is no time for such management. If we could track their previous medical records or ask the patient’s family, we may preliminarily understand the patient’s liver condition.

Patients with a history of hemodialysis had another 2 points added to their SAMIMS in this study. Previous studies have shown that kidney diseases are another factor closely related to postoperative survival, as a history of renal disorders or elevated creatinine levels were predictors of an increased risk of postoperative death [11,16,22]. Patients undergoing hemodialysis indicate a higher grade per the American Society of Anesthesiologists physical status classification system. Therefore, they have higher mortality rates after bowel resection for AMI compared with those without undergoing hemodialysis [6,47]. As such, the treatment strategy should not only focus on the early surgical intervention; proper fluid replacement and avoiding drug-induced renal toxicity are also crucial.

Regarding comorbidities, a history of congestive heart failure is considered an important negative prognosticator after four previous studies observed that patients with heart failure had higher mortality rates after surgery [12,22,24,25]; the investigators recommended that individuals being considered for surgery undergo careful preoperative cardiovascular workups as a result. However, in our study, myocardial infarction and perivascular disease were not observed to be independent preoperative risk factors, which was inconsistent with previous studies [12,21]; therefore, these variables were not incorporated into the SAMIMS. Conversely, the peptic ulcer was observed to be a negative prognosticator in our study even though it has not been previously reported. However, the relationship between these conditions is unclear. Moreover, why patients with AMI who have peptic ulcer exhibit higher mortality rates after bowel resection is poorly understood [29,48]; therefore, this topic requires further exploration. Cerebrovascular disease (1 point) and diabetes (1 point) were negative prognosticators. Previous studies have revealed that some comorbidities (such as diabetes and cerebrovascular diseases) in some patients impacted postoperative mortality rates [13,23]. However, there is no obvious explanation of why a history of diabetes and cerebrovascular disease are risk factors for short-term postoperative mortality of AMI.

Our study had several limitations. First, although the standard treatments of occlusive and non-occlusive have been established, in fact, in the management in different hospitals there remain some discrepancies because of different patients’ conditions, the equipment and the health care staff. Moreover, we could not confirm the etiologies of AMI However, the clinicians in our study select the patients with severe condition who are candidates for small bowel resection. Additionally, the hospitals with inadequate resources would not distinguish the etiologies of the AMI and treat the patients as standard guidelines. Therefore, it is necessary to understand the prognosis of AMI after bowel resection. We believe that there is research value in our scoring system. Second, due to unknown etiologies in AMI, using liver disease as a risk factor in predicting the mortality in AMI may lead to be a selection bias, since liver diseases are more important in mesenteric venous thrombosis than arterial AMI [49,50].

Third, the results of the secondary and administrative databases we used were restricted to the strict framework of Taiwan’s NHIRD without the possibility of conducting an in-depth analysis. For example, certain details such as demographic data, socioeconomic status, the timing of surgery, and details of each surgery were not recorded in the database and could not be obtained. Moreover, the study lacked some important clinical data, including vital signs, shock status, and laboratory findings, such as lactate levels. Previous studies observed that many other laboratory parameters could predict short-term death in patients with AMI after bowel resection. Forth, even though we performed specific correction testing while devising our scoring system, the discrimination remained poor. This limitation is common to every reported mortality prediction model involving rare clinical events and reflects the significant complexity in determining the causes of death within 30 days.

Fifth, our model may not be helpful for patients with unstable conditions or those thought to be ineligible for surgery. While a selection bias did exist, we included patients who were also diagnosed with mesenteric infarction and underwent bowel resection, as these conditions would be fully evaluated by surgeons from the outset. We aimed to generate a scoring system that would provide an additional metric for evaluating the mortality risk of these patients.

Sixth, there was a risk of miscoding since surgeons do not usually use the ICD-9; instead, they employ different systems and Health Insurance Surgical orders obtained from the Taiwanese NHI payment system. Moreover, we only used inpatient expenditure and admission data to identify patients and evaluate comorbidities; therefore, we may have underestimated the comorbidities of these patients. However, most ICD-9 codes during admission were assigned by professionals based on admission records, as these codes are directly related to payments from the NHI. Moreover, a table comparing ICD-9 and NHI payment system codes was generated by the NHI Administration at the Ministry of Health and Welfare, and we used relatively reliable data such as age, sex, date of admission, discharge condition, and death for our analyses. We also used the RCIPD, a relatively accurate database of comorbidities, given that specific confirmatory details such as surgical pathology or laboratory data are required for registration. As such, the odds of miscoding ought to be relatively low.

## 5. Conclusions

Surgery for acute mesenteric infarction is associated with high mortality. Although it is important to understand the mortality before surgery, only a few studies have involved scoring systems for predicting mortality following bowel resection for AMI with a main limitation due to the small patient numbers. We established a scoring system, the ‘Surgery for acute mesenteric infarction mortality score’ (SAMIMS), from a national database to evaluate the risks of 30-day hospital mortality among patients who experienced mesenteric infarction and underwent bowel resection. Our study analyzed data from more than 4000 patients. It is easy to apply preoperatively based patient’s age and underlying comorbidities in daily practice. Moreover, no matter whether in the derivation group or the validation group, SAMIMS can offer significant predictive value and enhance risk assessment before surgery. We believe the prediction model can offer evidence from a relatively large population and is easily accessible to assist clinicians in predicting surgical mortality outcome in treating acute mesenteric infarction.

## Figures and Tables

**Figure 1 jcm-11-05937-f001:**
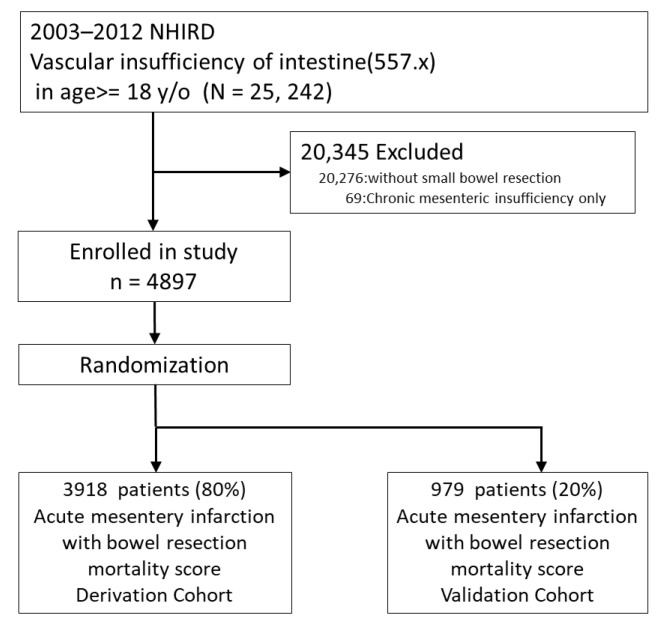
Flowchart of the study.

**Figure 2 jcm-11-05937-f002:**
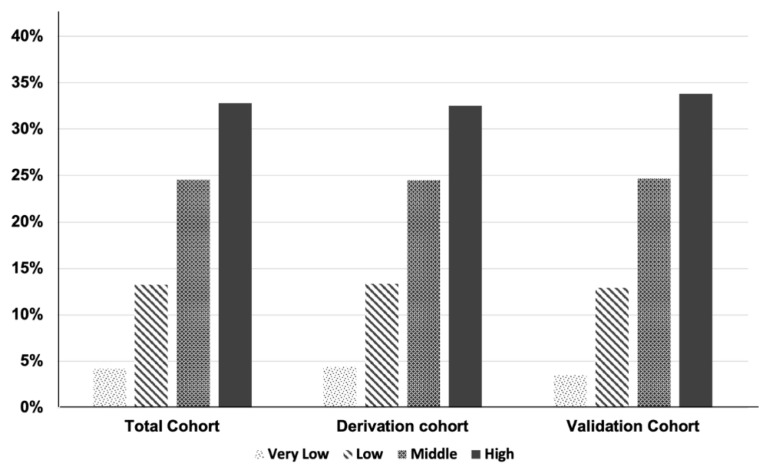
Thirty-day surgical mortality rates in the derivation and validation groups were categorized by surgery for acute mesenteric infarction mortality score (SAMIMS). Very low risk, SAMIMS = 0; low risk, SAMIMS = 1–3; intermediate risk, SAMIMS = 4–6; high risk, SAMIMS = 7–13).

**Table 1 jcm-11-05937-t001:** Basic characteristics of patients in the derivation and validation groups.

Variable	Derivation Group		Validation Group		*p*-Value
	*n*	3918	*n*	979	
Age in years, median (IQR)	71.92	(22.33)	72.00	(23.33)	0.838
Sex					0.067
Woman	1809	46.17%	420	42.90%	
Man	2109	53.83%	559	57.10%	
Major coexisting disease					
Myocardial infarction	242	6.18%	61	6.23%	0.941
Congestive heart failure	687	17.53%	162	16.55%	0.479
Vascular disease	2283	58.27%	559	57.10%	0.515
Cerebrovascular disease	787	20.09%	197	20.12%	1.000
Dementia	80	2.04%	30	3.06%	0.069
Chronic pulmonary disease	685	17.48%	169	17.26%	0.888
Rheumatic disease	63	1.61%	10	1.02%	0.237
Peptic ulcer disease	1028	26.24%	269	27.48%	0.442
Severe liver disease	93	2.37%	33	3.37%	0.090
Diabetes	1039	26.52%	255	26.05%	0.777
Hemiplegia	154	3.93%	38	3.88%	1.000
Under dialysis	354	9.04%	93	9.50%	0.664
Malignancy	687	17.53%	187	19.10%	0.263
Heart conduct disease	801	20.44%	206	21.04%	0.691
Hypertension	1869	47.70%	441	45.05%	0.142
Hyperlipidemia	405	10.34%	105	10.73%	0.726
Gout	326	8.32%	97	9.91%	0.127
Obesity	8	0.20%	2	0.20%	1.000
Laparoscope operation	98	2.5%	23	2.35%	0.908
Perioperative outcomes					
30-day mortality	684	17.46%	168	17.16%	0.851
Median LOH (IQR)	16	(18)	16	(19)	0.318

There were no significant differences between the groups regarding age, sex, or any of the covary-ates. LOH: length of hospital stay; IQR: interquartile range.

**Table 2 jcm-11-05937-t002:** ‘Surgery for acute mesenteric infarction mortality score’ system.

Age	Score
Age > 62	3
Comorbidities	Score
Severe liver disease	4
Hemodialysis	2
Congestive heart failure	1
Peptic ulcer disease	1
Cerebrovascular disease	1
Diabetes	1

All variables show statistical significance for predicting 30-day mortality in multivariable logistic regression. The regression coefficient of variables significantly related to mortality (*p* < 0.05) was multiplied by numbers and rounded to the nearest integer to create a score on an additive scale. Then, this score was applied to the derivation and validation group for further evaluation.

**Table 3 jcm-11-05937-t003:** Risk of 30-day mortality in the derivation and validation groups.

Derivation Group				Validation Cohort			
Score	Mortality Rate	OR	*p*	Score	Mortality Rate	OR	*p*
0 (Very Low)	4.4%	1		0 (Very Low)	3.5%	1	
1–3 (Low risk)	13.4%	3.332 (2.287–4.857)	<0.001	1–3 (Low risk)	12.9%	4.117 (1.809–9.369)	0.001
4–6 (Median risk)	24.5%	7.004 (4.902–10.008)	<0.001	4–6 (Median risk)	24.7%	9.082 (4.130–19.974)	<0.001
7–13 (High risk)	32.5%	10.410 (6.763–16.023)	<0.001	7–13 (High risk)	33.8%	14.165 (5.724–35.053)	<0.001
ROC: AUC = 0.677, *p* < 0.001, 95% CI: 0.660–0.689, Calibration (Hosmer–Lemeshow goodness-of-fit test) χ^2^ = 19.887 (*p* = 0.001)				ROC: AUC = 0.696, *p* < 0.001, 95% CI: 0.666–0.725, Calibration (Hosmer–Lemeshow goodness-of-fit test) χ^2^ = 5.067 (*p* = 0.280)			

Statistical significantly increased the risk of postoperative mortality with increasing preoperative scores demonstrated in 4 subgroups. OR, odds ratio; ROC, receiver operating characteristic curve; AUC, area under the curve; CI, confidence interval.

## Data Availability

This study is based in part on data from the National Health Insurance Research Database (NHIRD), provided by the National Health Insurance Administration of the Ministry of Health and Welfare and managed by the National Health Research Institutes (registration number NHIRD-103-246). The data utilized in this study cannot be made available in the manuscript, supplementary files, or in a public repository due to the “Personal Information Protection Act” executed by Taiwan’s government, which took effect in 2012. Data are available with the permission of NHIRD.

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
