# Peer review of "Mortality Prediction Model before Surgery for Acute Mesenteric Infarction: A Population-Based Study"

_jcm, 2022, doi:10.3390/jcm11195937_

Round 1

Reviewer 1 Report

Thank you very much for giving me the opportunity to review the manuscript entitled “Mortality Prediction Model Before Surgery for Acute Mesenteric Infarction: A Population-based Study”.  In this manuscript authors present a new 30-days mortality score (SAMIMS) which they developed on the basis of multivariate analysis of 3900 cases of patients with mesenteric infarction. The score reliability was afterwards tested on almost 1000 cases. The score is built on six identified major comorbidities and the age of the patient and allows prediction of postoperative 30-days mortality, preoperatively.    

All data were provided by the National Health Insurance Research Database (NHIRD) of Taiwan according to the study protocol, which was approved by the Institutional Review Board.  

The statistical analyses is very well done as well as the construction of the score system.  

In general, the authors present a well-done paper with a novel score system.

General concept comments:

Article: The authors mention that there are many different preoperative models/scores for a postoperative mortality prediction. So I would recommend to compare the new score system (SAMIMS) with one or two of the validated scores (e.g. POSPOM) to underline the reliability of SAMIMS.  

Author Response

We also upload a certificate for the english editing.

Thanks you very much.

Reviewer 2 Report

In this paper the authors defined a new risk prediction score for surgery for acute mesenteric infarction by using national registry database. Although the study seems to have been thoroughly done, there are serious limitations in the methods. 

Followings are specific comments.

1. The method used for randomization of the patients needs to be described.

2. Cerebral disease and severe liver disease are vague category. What were the definitions of these conditions?

3. The cutoff of age (62 years old) and weighting of the score, i.e., 3 points for age, 4 points for liver disease, and so on, seems quite arbitrary. I can presume these were defined based on the multivariate analysis, but the rationale needs to be explained in the manuscript.

4. The biggest limitation of this study is that the national database they used only provides expenditure and admission data. It seems impossible to differentiate past history and concomitant condition from the database. Patients in severe condition due to AMI can develop congenital heart failure and require hemodialysis, and it’s obvious these patients have high risk of mortality. Therefore, it is important to distinguish whether these conditions existed before AMI or were results from AMI to create a risk prediction score.

5. The name of the scoring is inconsistent, i.e., SAMIMS or AMIMS. The authors should be careful to use the term they defined.

Author Response

Thanks you very much.

Round 2

Reviewer 2 Report

The questiones were adequetly addressed. The manuscript is better now.

Author Response

Response to Reviewer 2 Comments

Thanks for your reviewing and valuable opinions.